# Human Peripheral Blood-Derived Endothelial Colony-Forming Cells Are Highly Similar to Mature Vascular Endothelial Cells yet Demonstrate a Transitional Transcriptomic Signature

**DOI:** 10.3390/cells9040876

**Published:** 2020-04-03

**Authors:** Anton G. Kutikhin, Alexey E. Tupikin, Vera G. Matveeva, Daria K. Shishkova, Larisa V. Antonova, Marsel R. Kabilov, Elena A. Velikanova

**Affiliations:** 1Research Institute for Complex Issues of Cardiovascular Diseases, 6 Sosnovy Boulevard, Kemerovo 650002, Russia; matveeva_vg@mail.ru (V.G.M.); shidk@kemcardio.ru (D.K.S.); antonova.la@mail.ru (L.V.A.); telella@mail.ru (E.A.V.); 2Institute of Chemical Biology and Fundamental Medicine, Siberian Branch of the Russian Academy of Sciences, 8 Lavrentiev Avenue, Novosibirsk 630090, Russia; alenare@niboch.nsc.ru (A.E.T.); kabilov@niboch.nsc.ru (M.R.K.)

**Keywords:** endothelial colony-forming cells, peripheral blood mononuclear cells, coronary artery endothelial cells, umbilical vein endothelial cells, adipose tissue-derived stromal vascular fraction, endothelial lineages, gene expression, RNA-seq, transcriptome profiling, transcriptomic signatures

## Abstract

Endothelial colony-forming cells (ECFC) are currently considered as a promising cell population for the pre-endothelialization or pre-vascularization of tissue-engineered constructs, including small-diameter biodegradable vascular grafts. However, the extent of heterogeneity between ECFC and mature vascular endothelial cells (EC) is unclear. Here, we performed a transcriptome-wide study to compare gene expression profiles of ECFC, human coronary artery endothelial cells (HCAEC), and human umbilical vein endothelial cells (HUVEC). Characterization of the abovementioned cell populations was carried out by immunophenotyping, tube formation assay, and evaluation of proliferation capability while global gene expression profiling was conducted by means of RNA-seq. ECFC were similar to HUVEC in terms of immunophenotype (CD31^+^vWF^+^KDR^+^CD146^+^CD34^-^CD133^-^CD45^-^CD90^-^) and tube formation activity yet had expectedly higher proliferative potential. HCAEC and HUVEC were generally similar to ECFC with regards to their global gene expression profile; nevertheless, ECFC overexpressed specific markers of all endothelial lineages (*NRP2*, *NOTCH4*, *LYVE1*), in particular lymphatic EC (*LYVE1*), and had upregulated extracellular matrix and basement membrane genes (*COL1A1*, *COL1A2*, *COL4A1*, *COL4A2*). Proteomic profiling for endothelial lineage markers and angiogenic molecules generally confirmed RNA-seq results, indicating ECFC as an intermediate population between HCAEC and HUVEC. Therefore, gene expression profile and behavior of ECFC suggest their potential to be applied for a pre-endothelialization of bioartificial vascular grafts, whereas in terms of endothelial hierarchy they differ from HCAEC and HUVEC, having a transitional phenotype.

## 1. Introduction

Coronary artery bypass graft surgery remains an efficient and widespread surgical option to treat coronary artery disease [1], yet availability of autologous conduits such as internal mammary arteries or saphenous veins is often limited in patients with widespread atherosclerotic vascular disease or in those whose vessels are anatomically incompatible or have already been harvested for a previous procedure [2,3,4]. Allogeneic and xenogeneic blood vessels demonstrate only a limited efficiency due to the risk of transmissible disease, graft-versus-host disease, infection, and calcification [4,5] while prosthetic grafts fabricated from biostable synthetic polymers such as poly(ethylene terephthalate), expanded poly(tetrafluoroethylene), and polyurethanes show inferior patency rates in small-diameter applications due to poor endothelialization, low blood flow, and compliance mismatch, all resulting in intimal hyperplasia, thrombosis, or (pseudo)aneurysms [2,3,4,5]. In addition, both xenogeneic and biostable synthetic vascular conduits lack growth adaptation potential and therefore demand repeated surgery and ultimately lead to unacceptable long-term outcomes [5,6]. Hence, vascular tissue engineering has emerged as one of the most promising approaches for producing mechanically competent and biocompatible small-diameter vascular substitutes [2,4,7,8]. This frequently involves the use of biodegradable polymers to provide a tubular scaffold for cell adhesion, proliferation, and intramural migration followed by the de novo formation of the vascular tissue and subsequent scaffold degradation [8].

The major advantage of the biodegradable vascular grafts over biostable ones is better endothelialization, a process pivotal for the prevention of thrombotic occlusion [9,10,11]. However, thrombosis generally occurs within the first minutes or hours upon contact of the blood with the polymer surface, negating improved biocompatibility of biodegradable polymers. Hence, pre-endothelialization is strongly required to secure zero thrombogenicity of the vascular graft and screening of pertinent endothelial cell sources is rapidly ongoing. Recent studies report successful differentiation of peripheral blood mononuclear cells (PBMC) into endothelial colony-forming cells (ECFC) [12,13] which is particularly efficient after percutaneous coronary intervention due to a mechanical injury provoking the release of ECFC precursors into the bloodstream [14]. ECFC exhibit high angiogenic [15,16] and proliferative [16] capability and can be suggested as an appropriate cell population for the endothelialization of the tubular scaffolds prior to implantation.

In contrast to short-term patency which is largely determined by endothelial integrity, long-term patency depends on endothelial homeostasis maintained by a proper gene expression profile. Distinct endothelial lineages have considerable differences regarding their transcriptome, and the extent of endothelial cell heterogeneity is currently a matter of debate [17,18,19]. Nevertheless, endothelial identity is mandatory to control vascular tone and for the paracrine regulation of vascular specification within the graft [20,21,22]. Accordingly, transcriptomic signatures of pre-seeded endothelial cells (EC) should correspond to those of mature resident EC.

To assess whether ECFC are suitable for in vitro pre-endothelialization of vascular grafts, we compared the global gene expression profile of human PBMC-derived ECFC in comparison with human coronary artery endothelial cells (HCAEC), human umbilical vein endothelial cells (HUVEC), and human subcutaneous adipose tissue-derived stromal vascular fraction (SAT-SVF) by means of RNA-sequencing (RNA-seq). We found that the baseline gene expression profile of ECFC is close to that of HCAEC and HUVEC but expectedly different from SAT-SVF, testifying to their utility for the seeding of tubular scaffolds before implantation to improve their short- and long-term performance.

## 2. Materials and Methods

### 2.1. Cell Culture

Peripheral blood (20 mL) was withdrawn from 8 male patients during percutaneous coronary intervention performed in the Research Institute for Complex Issues of Cardiovascular Diseases (Kemerovo, Russian Federation). The study design was approved by the Local Ethical Committee (ID 657460, approved 28 October 2016). All patients provided written informed consent before the recruitment after receiving a full explanation of the study. A complete blood count was performed utilizing an automated hematology analyzer (MEK8222K, Nihon Kohden, Tokyo, Japan). Average white blood cell count was 6.8 × 10^9^/L (13.6 × 10^7^ in 20 mL). PBMC were isolated using Histopaque density media 1077 (10771, Sigma, St. Louis, MO, USA) according to the manufacturer’s instructions. The cells obtained from the interphase were washed twice with phosphate buffered saline (PBS, 70011044, Thermo Fisher Scientific, MA, USA) followed by centrifugation. We performed isolation and enrichment of ECFC using the modified protocol by Kolbe et al. [23]. Briefly, cells (2.8 × 10^7^ in total) were resuspended in the EGM-2MV (CC-3202, Lonza, Basel, Switzerland) medium supplemented with 5% fetal bovine serum (FBS) (SH3007103, HyClone) and seeded into the collagen-coated 25 cm^2^ flasks (356484, Corning, New York, NY, USA). In the first two days, the medium was changed daily to remove non-adherent cells and debris, and then was changed three times a week. After one week of culture, the cells were dissociated with trypsin and reseeded to fibronectin-coated 75 cm^2^ flasks (354521, Corning). The frequency of ECFC at the time of reseeding was 1 per 10^7^ cultured PBMC (3 or 4 colonies per 75 cm^2^ flask). Passaging was performed at 70%–80% confluence. Immunophenotyping and functional assays were performed at the 19th–22nd day of culture (average ECFC yield 1.5 × 10^6^–2.0 × 10^6^ cells per 75 cm^2^ flask). Cells were counted after the trypsinization using an automated cell counter (LUNA-II, Logos Biosystems, Seoul, South Korea).

SAT (1.7–2.0 cm^3^) was harvested from the same patients, washed in Hank’s balanced salt solution (HBSS, 14185052, Gibco), minced, and treated with 0.2% collagenase type I (17100017, Gibco) during 30 minutes at 37 °C. Enzymatic digestion was stopped by 10% FBS. Dissociated tissue was then filtered (100 µm) and centrifuged at 1000× *g*. Cell pellet was resuspended in EGM-2MV, seeded into collagen-coated 25 cm^2^ flasks (354484, Corning) and then cultured as above.

HUVEC were isolated from umbilical cords according to the modified protocol of Jaffe et al. [24], while primary HCAEC were purchased from Cell Applications (San Diego, CA, USA) (300K-05a). For the culture of HUVEC and HCAEC, we used EGM-2MV (CC-3202, Lonza) medium supplemented with 5% FBS (SH3007103, HyClone) similar to the culture of ECFC and SAT-SVF. Initially, we seeded 0.5 × 10^6^ cells per collagen-coated 25 cm^2^ flask (356484, Corning). At the next passage, HUVEC and HCAEC were reseeded into fibronectin-coated 75 cm^2^ flasks (354521, Corning). For all analyses, we collected HUVEC and HCAEC of the same passage and in the similar amount as ECFC and SAT-SVF (passage 3 or 4, average yield 1.5 × 10^6^–2.0 × 10^6^ cells per 75 cm^2^ flask).

All cell lines were cultured at standard cell culture conditions, and cell culture procedures were carried out under strict aseptic conditions. Visual control of culture growth was performed daily.

### 2.2. Immunophenotyping

A total of 100 μL containing 0.5 × 10^5^–1.0 × 10^5^ ECFC, HUVEC, or SAT-SVF removed from the plates and washed with PBS were taken for the staining with conjugated monoclonal antibodies against human antigens: (1) allophycocyanin (APC)-conjugated anti-CD31 (303116, BioLegend, San Diego, CA, USA, 1:60 dilution); (2) fluorescein isothiocyanate (FITC)-conjugated anti-von Willebrand factor (vWF, ab8822, Abcam, Cambridge, UK, 1:100); (3) phycoerythrin (PE)-conjugated anti-kinase insert domain receptor (KDR, 560494, BD Biosciences, San Jose, CA, USA, 1:50); (4) phycoerythrin-cyanine 7 (PE/Cy7)-conjugated anti-CD146 (361008, BioLegend, 1:60); (first panel); (5) FITC-conjugated anti-CD34 (343504, BioLegend, 1:60); (6) APC-conjugated anti-CD133 (372806, BioLegend, 1:60); (7) Krome Orange (KrO)-conjugated anti-CD45 (A96416, Beckman Coulter, 1:50); (second panel); (8) FITC-conjugated anti-CD90 (328108, BioLegend, 1:60); (9) APC/Cy7-conjugated anti-CD73 (344022, BioLegend, 1:60) (third panel). Fixation and permeabilization of the cells were performed using the IntraPrep Permeabilization Reagent (A07803, Beckman Coulter, Brea, CA, USA) when staining intracellular vWF. Samples were then resuspended in PBS and analyzed on a CytoFlex flow cytometer with CytExpert software (Beckman Coulter).

To verify flow cytometry results, the cells were cultured on fibronectin-coated coverslips and stained for characteristic endothelial markers CD31 and vWF. Briefly, the cells were fixed for 10 min in 4% paraformaldehyde (P6148, Sigma). To stain intracellular markers (vWF), permeabilization of the cells was additionally performed by treatment with 0.1% Triton X-100 (T8787, Sigma) for 15 min. To block non-specific binding, the cells were treated with 1% BSA (A9418, Sigma) for 1 h. Samples were then incubated with primary antibodies (anti-CD31, ab119339, Abcam, 1:100) overnight at 4 °C and further with secondary antibodies (Alexa Fluor 568-conjugated goat anti-mouse, ab175473, Abcam, 1:600) and sheep FITC-conjugated anti-vWF antibody (ab8822, Abcam, 1:100). Control samples were mock-stained with 1% BSA instead of primary antibodies with all other procedures according to the protocol. Nuclei were counterstained with 4′,6-diamidino-2-phenylindole (DAPI) (D9542, Sigma) at a concentration of 10 μg/mL. Samples were mounted using ProLong Gold Antifade (P36930, Invitrogen) medium. Examination was carried out by confocal microscopy (LSM 700, Carl Zeiss, Oberkochen, Germany).

### 2.3. Functional Assays

For the evaluation of ECFC, HUVEC, and SAT-SVF tube-forming capacity, 200 μL Matrigel (354234, Corning) was added to each well of a 24-well plate for 30 min at room temperature to allow polymerization. Cells were seeded at 1.0 × 10^5^ cells per well and resuspended in EGM-2MV supplemented with 5% FBS. The formation of capillary-like tubes was monitored after 16 h by phase contrast microscopy (AxioObserver.A1, Carl Zeiss). Semi-quantitative image analysis (5 representative images per cell line) was performed employing WimTube (Wimasis, Cordoba, Spain).

To assess acetylated low-density lipoprotein cholesterol (acLDL) uptake and *Ulex Europaeus* agglutinin 1 (UEA) binding by ECFC and HUVEC, a total of 2.4 μg/mL 1,1′-dioctadecyl-3,3,3′,3′-tetramethylindocarbocyanine perchlorate (DiI)-labeled acLDL (L3484, Invitrogen) was added to the cells with the following incubation for 2 h at 37 °C. The cells were then fixed with 2% paraformaldehyde for 15 min and incubated for 1 h with FITC-conjugated UEA (L9006, Sigma) at a concentration of 10 μg/mL. Nuclei were counterstained with DAPI (1.5 μg/mL). Samples were mounted (ProLong Gold Antifade) and assessed utilizing confocal microscopy (10 representative fields of view per cell line).

Fluorescent staining of dividing nuclei in ECFC and HUVEC was performed using the Click-iT Plus EdU Alexa Fluor 488 Imaging Kit (C10637, Invitrogen) according to the manufacturer’s protocol. The EdU exposure time was 6 h. The samples were counterstained with DAPI (10 μg/mL), mounted (ProLong Gold Antifade), and analyzed using confocal microscopy. Positive cells (green) were counted in 10 representative fields of view, and then their ratio to the total number of cells in these fields was calculated.

The proliferative activity of ECFC and HUVEC (passage 4) was additionally evaluated by non-invasive electrical impedance monitoring (xCELLigence Real-Time Cell Analyzer Dual-Plate, ACEA Biosciences). Cells were seeded into 16-well E-plates (2801032, ACEA Biosciences, 2 × 10^4^ cells per well) in duplicate, and impedance was measured over 100 h. Cell-free culture medium was applied as a blank. Proliferation capability was defined as cell index doubling time calculated automatically by the instrument software.

### 2.4. RNA-Seq

RNA-seq was performed in SB RAS Genomics Core Facility (ICBFM SB RAS, Novosibirsk). Upon the withdrawal of culture medium and washing in ice-cold phosphate buffered saline (four 75 cm^2^ flasks per group), cells were lysed with TRIzol (15596018, Invitrogen) with the following total RNA isolation (Purelink RNA Micro Scale Kit, 12183016, Invitrogen) and DNAse treatment (DNASE70, Sigma). RNA integrity index (RIN) was assessed using RNA 6000 Pico Kit (5067-1513, Agilent) and Bioanalyzer 2100 (Agilent) (Appendix A) while RNA quantification was carried out using NanoDrop 2000 (Thermo Scientific) and Qubit 4 (Invitrogen). For the 1 µg of isolated RNA, we performed rRNA depletion (RiboCop rRNA Depletion Kit V1.2, 037.96, Lexogen) followed by DNA library preparation (SENSE Total RNA-Seq Library Prep Kit, 042.96, Lexogen) and quality control (High Sensitivity DNA Kit, 5067-4626, Agilent) (Appendix A). DNA libraries were then quantified by qPCR (CFX96 Touch, Bio-Rad), pooled in equimolar amounts and sequenced (HiSeq 2000, Illumina) using 2 × 132 bp chemistry.

SENSE Total RNA-Seq uses a 9-nt-long random sequence of the starter and a 6-nt-long random sequence of the stopper hybridized to the RNA template. Therefore, it removed the first nine nucleotides from read 1 and the first six nucleotides from read 2 by cutadapt v.1.18. After the filtration of cut reads by quality (QV > 20) and length (> 20), we performed an adapter trimming by TrimGalore v.0.4.4. The average number of reads exceeded 10,000,000. Read mapping to the human genome (hg38 with Ensembl annotation v.38.93) was conducted using CLC GW 12.0 (Qiagen) according to the following parameters: similarity fraction = 0.8, length fraction = 0.8, mismatch cost = 2, insertion cost = 3, deletion cost = 3 (Appendix A). To define differentially expressed genes (DEGs), we used multifactorial statistical analysis (CLC GW 12.0) based on the negative binomial regression.

The data reported in this study have been deposited in NCBI’s Gene Expression Omnibus and are accessible through GEO Series accession number GSE131995 (https://www.ncbi.nlm.nih.gov/geo/query/acc.cgi?acc=GSE131995).

### 2.5. Proteomic Profiling

Validation of RNA-seq was performed by means of dot blotting and conventional Western blotting. Upon the withdrawal of culture medium and washing in ice-cold phosphate buffered saline (two 75 cm^2^ flasks per group), cells were lysed with RIPA buffer (89901, Thermo Scientific), and total protein concentration was measured using Pierce BCA Protein Assay Kit (23227, Thermo Scientific). Protein samples (15 µg per sample) were mixed with NuPAGE LDS Sample Buffer (NP0008, Invitrogen) and NuPAGE Sample Reducing Agent (NP0004, Invitrogen), denatured at 99 °C during 5 min, loaded into 10-well NuPAGE 4–12% Bis-Tris Protein Gels of 1.5 mm thickness (NP0335BOX, Invitrogen) and separated in NuPAGE MES SDS Running Buffer (NP000202, Invitrogen) containing NuPAGE Antioxidant (NP0005, Invitrogen) at 150 V for 1.5 h. A 1:1 mixture of Novex Sharp Pre-stained Protein Standard (LC5800, Invitrogen) mixed with MagicMark XP Western Protein Standard (LC5602, Invitrogen) was utilized as a molecular-weight size marker. Dry protein transfer was conducted during a standard 7-minute protocol employing polyvinylidene fluoride iBlot 2 Transfer Stacks (IB24001, Invitrogen) and iBlot 2 Gel Transfer Device (Invitrogen). Membranes were then incubated in iBind Flex blocking solution (SLF2020 × 4, Invitrogen) for 2 h and then processed in iBind Western Device (Invitrogen) overnight with recommended primary and secondary antibody dilutions (iBind Flex solution). The following primary antibodies was used: mouse anti-CD31 (ab9498, Abcam, 1:1000), rabbit anti-VE-cadherin (36-1900, Invitrogen, 1:100), rabbit anti-vWF (ab6994, Abcam, 1:500), rabbit anti-VEGFR2/KDR (ab39256, Abcam, 1:1000), rabbit anti-CD34 (ab81289, Abcam, 1:1000), rabbit anti-neuropilin 1 (ab81321, Abcam, 1:1000), rabbit anti-HEY2 (ab221931, Abcam, 1:200), mouse anti-NR2F2/COUP-TFII (ab41859, Abcam, 1:200), rabbit anti-LYVE1 (ab14917, Abcam, 1:200), rabbit anti-VEGFR3 (ab27278, Abcam, 1:100), rabbit anti-Snail + Slug (ab180714, Abcam, 1:500), mouse anti-N-cadherin (MA5-15633, Invitrogen, 1:500), and goat anti-β-tubulin (loading control, ab21057, Abcam, 1:1000). Horseradish peroxidase-conjugated goat anti-mouse (AP130P, Sigma-Aldrich, 1:1000), goat anti-rabbit (7074S, Cell Signaling Technology, 1:200), or donkey anti-goat (ab205723, Abcam, 1:400) were used as secondary antibodies. Chemiluminescence detection was carried out by incubation of the membrane in SuperSignal West Pico PLUS Chemiluminescent Substrate (34580, Thermo Scientific) for 1 min followed by 12-minute exposure in C-DiGit Blot Scanner (LI-COR Biosciences). Dot blotting against 55 angiogenesis-related proteins was performed using Proteome Profiler Human Angiogenesis Array Kit (ARY007, R&D Systems, Minneapolis, MN, USA) according to the manufacturer’s instructions (275 µg of cell lysate per sample was loaded onto the membrane).

### 2.6. Statistical Analysis

Statistical analysis was performed using GraphPad Prism 8 (GraphPad Software, San Diego, CA, USA). Data were represented by the median, 25th and 75th percentiles, and range. Groups were compared by Mann–Whitney U test. *P* values ≤ 0.05 were regarded as statistically significant.

Statistically significant DEGs were defined as those with fold change ≥ 2 and false discovery rate (FDR) *p*-value < 0.05. In Gene Ontology (GO) functional enrichment analysis (CLC GW 12.0), we considered only statistically significant (FDR *p*-value < 0.05) categories with the ratio of DEGs to total genes exceeding 0.5 (50%).

## 3. Results

### 3.1. Differentiation of ECFC from PBMC

To differentiate ECFC from PBMC, we applied the protocol established by Kolbe et al. [23], including culture of PBMC in EGM-2MV medium in collagen-coated flasks with the further reseeding of the intermediate cell population into fibronectin-coated flasks and finally into the standard culture dishes. Both flow cytometry (Figure 1A, Table 1) and confocal microscopy (Figure 1B) confirmed endothelial specification in terms of CD31, vWF, KDR, and CD146 positive staining and CD45-negative staining. ECFC formed capillary-like tubes in Matrigel (Figure 1C) with the characteristics similar to HUVEC (Figure 1D), internalized acLDL with the concurrent UEA binding (Figure 2A), and exhibited a remarkable proliferation capability superior to HUVEC (Figure 2B–D), thereby fulfilling requirements for endothelial identity.

### 3.2. ECFC and SAT-SVF Have Remarkable Differences in Global Gene Expression Profile

For the comprehensive and unbiased evaluation of the baseline gene expression profile in ECFC as compared with HCAEC and HUVEC, we performed RNA-seq of indicated cell cultures, suggesting SAT-SVF as a negative control population according to the immunophenotyping and functional analyses (Appendix A). A total of 2696 DEGs were identified between ECFC and SAT-SVF (Appendix A); hierarchical clustering by the transcription level of top DEGs revealed a clearly defined aggregation of RNA samples from SAT-SVF and ECFC into separate clusters (Appendix A). These results have been further confirmed by a principal component analysis (Appendix A).

### 3.3. ECFC Demonstrate High Similarity to Resident Vascular EC yet Represent a Transitional Cell Population

We next compared the baseline gene expression profile of ECFC to HCAEC (Appendix A) and HUVEC (Appendix A). Hierarchical clustering demonstrated certain transcriptomic signatures for both of these comparisons (Figure 3). When ECFC were compared with HCAEC, gene set enrichment analysis revealed significant differences in gene expression profile exclusively in vascular endothelial growth factor-activated receptor activity (Appendix A). However, it neither detected other statistically significant differences between ECFC and HCAEC GO categories nor observed any differences between ECFC and HUVEC. With the aim to pinpoint DEGs between ECFC, HCAEC, and HUVEC, we manually annotated them regarding their role in endothelial biology.

According to such annotation, when compared with HCAEC, ECFC had a higher expression of pan-endothelial markers *KDR* (2.2-fold) and *VWF* (3.9-fold), endothelial progenitor cell marker *CD34* (23.9-fold), venous endothelial specification marker *NRP2* (7.9-fold), lymphatic endothelial lineage markers *FLT4* (11.6-fold) and *LYVE1* (45.7-fold), subunits of collagen type IV composing endothelial basement membrane (*COL4A1* and *COL4A2*, 2.5 and 3-fold, respectively), and subunits of collagen type I, a major component of the extracellular matrix (*COL1A1* and *COL1A2*, 926-fold and 43.5-fold, respectively). In contrast, HCAEC had higher expression of the gene encoding key Notch pathway transcription factor *HEY2* (523-fold), *NOS3* gene (4.9-fold), and *FLT1* gene (encodes VEGFR1, 3.9-fold). Comparison in terms of manually annotated categories found that HCAEC overexpressed genes encoding proteins responsible for vascular development (5 versus 0 in ECFC) while ECFC overexpressed genes providing ECM production/remodeling (13 versus 7 in HCAEC) and endothelial integrity (15 versus 8 in HCAEC); other categories did not show any notable differences (Figure 4A).

In comparison with HUVEC, ECFC had higher expression of *NOTCH4* gene (3.1-fold), Notch ligand *DLL2* (4.1-fold), *LYVE1* (18.7-fold), and *COL1A1* (914-fold), while HUVEC had higher expressions of the key endothelial mechanotransduction transcription factor *KLF4* (45-fold), mesenchymal marker *CDH2* (encodes N-cadherin, 7.6-fold), and *VEGFA* (5-fold). Comparison of ECFC to HUVEC regarding manually annotated categories demonstrated that HUVEC overexpressed genes encoding proteins stimulating migration of EC (4 versus 0 in ECFC) and angiogenesis (19 versus 6 in ECFC); other categories displayed no appreciable differences (Figure 4B).

As compared with other cells of endothelial identity, ECFC were characterized by a higher baseline expression of the genes responsible for basement membrane composition (*COL4A1* and *COL4A2* encoding collagen type IV) and extracellular matrix (*COL1A1* and *COL1A2* encoding collagen type I) and expressed markers of all three endothelial lineages (i.e. arterial, venous, and lymphatic specification), thereby showing a transitional transcriptomic signature. Functional enrichment of the GO-defined gene networks responsible for endothelial homeostasis and angiogenesis (Appendix A) demonstrated a higher similarity to HUVEC (32 DEGs out of 653 included into this subanalysis) than to HCAEC (87/653), albeit HCAEC and HUVEC differed more significantly (112/653) than each of these populations compared with ECFC (Appendix A and Figure 5). In addition to functional enrichment analysis, a Venn diagram (Figure 6) and principal component analysis (Figure 7), which included all DEGs, also indicated that the gene expression profile of ECFC is closer to HUVEC (261 DEGs) than to HCAEC (470 DEGs); nevertheless, differences between HCAEC and HUVEC were not more pronounced when all DEGs were included into the analysis (420 DEGs). Analysis of manually annotated categories found comparable differences in both number of differentially expressed categories and DEGs within them between ECFC, HCAEC, and HUVEC (two categories per ECFC–HCAEC and ECFC–HUVEC comparisons including 28 and 29 DEGs, respectively). Manual screening of pluripotency markers documented only scarce differences between ECFC, HCAEC, and HUVEC, but the differences between HCAEC and HUVEC (7 DEGs) were higher than in pairwise comparison of ECFC with either HCAEC (5 DEGs) or HUVEC (4 DEGs) (Figure 8). Taken together, these results suggested intermediate specification of ECFC in relation to HCAEC and HUVEC.

### 3.4. Proteomic Profiling Validates Transcriptome-Wide Comparison Confirming Intermediate Endothelial Hierarchy of ECFC

To verify the key results obtained by RNA-seq, we measured the relative levels of endothelial lineage markers and angiogenic molecules across SAT-SVF, ECFC, HCAEC, and HUVEC utilizing different Western blotting modalities. Indeed, abundant expression of angiopoietin 2 (Ang-2) and endothelin-1 (ET-1) has been detected in the lysate of ECFC but not SAT-SVF in accordance with the fold change values acquired from RNA-seq (1600 and 817, respectively); however, the reverse was the case for tissue inhibitor of metalloproteinases 1 (TIMP-1, fold change −10, Appendix A).

Similar to RNA-seq, proteomic profiling did not find major differences across the EC, yet ECFC generally had either the highest or lowest expression of angiogenic molecules in head-to-head comparison with HCAEC and HUVEC (Figure 9A). Analysis of pan-endothelial markers verified the endothelial phenotype of ECFC which had high levels of CD31, VE-cadherin, VEGFR2/KDR, CD34, and neuropilin-1 (NRP1) (Figure 9B). Blotting against endothelial lineage and endothelial-to-mesenchymal transition markers testified to the transitional phenotype of ECFC in terms of intermediate HEY2, LYVE1, VEGFR3, Snail, and Slug content when compared to HCAEC and HUVEC, which overexpressed HEY2 and COUP-TFII, respectively (Figure 9B).

## 4. Discussion

ECFC represent a cell population that can be differentiated from peripheral blood [12,13,25], umbilical cord blood [25,26,27,28], placenta [29,30,31], bone marrow [32], white adipose tissue [33], peripheral lung tissue [34] and induced pluripotent or embryonic stem cells [35], and is characterized by an endothelial immunophenotype (CD31+vWF+KDR+CD146+CD45-), intracellular CD133 expression, acLDL uptake and UEA binding, considerable tube-forming capacity and high proliferation capability [16,25,28]. The combination of endothelial identity with high proliferative and neovascularization potential suggests ECFC as an efficient tool for regenerative medicine applications including pre-endothelialization [36] and pre-vascularization [27,37,38,39] of tissue-engineered constructs. However, usefulness of ECFC for indicated tasks depends to a large extent on their similarity to resident EC, such as HCAEC, that are particularly important for pre-seeding of small-diameter vascular grafts, or HUVEC which represent the far most frequently utilized endothelial cell line in biomaterial research [17]. As phenotypic features are tightly regulated by the gene expression networks, we sought to investigate the transcriptome profile of ECFC, HCAEC, and HUVEC in comparison with conceivably different SAT-SVF.

We found major differences in gene networks responsible for VEGF signaling, proliferation, migration, organization of intercellular junctions, and neovascularization between ECFC and SAT-SVF. Further, expression of endothelial signature genes (*PECAM1*, *KDR*, *CDH5*, *FLT1*, *VWF*, *TIE1*, *CD34*, *NOTCH1*, *NOTCH4*, *CXCR4*, *FLI1*, *MYO5C*, *ANGPT2*), basement membrane genes (*COL4A1*/*COL4A2*), and *NOS3* gene encoding endothelial nitric oxide synthase, a crucial enzyme for maintenance of endothelial homeostasis, was significantly increased in ECFC. Taken together with a pronounced tube-forming activity of ECFC in a 3D culture model (Matrigel) and distinct immunophenotype, these findings may indicate a higher capability of CD31+vWF+KDR+CD146+ ECFC to assemble blood vessels as compared with CD31-vWF-KDR-CD146- SAT-SVF. In contrast to our results, a similar study identified minor differences between immunophenotype, tube-forming capacity, and global gene expression profiles (223 DEGs compared with 2696 in our study) of ECFC and SAT-SVF-derived EC, possibly due to the additional enrichment of SVF by removal of CD44+ cells [40].

Expectedly, ECFC were substantially more similar to HCAEC (470 DEGs) and HUVEC (261 DEG) than to SAT-SVF (2696 DEGs). The extent of heterogeneity between HCAEC and HUVEC (420 DEGs) generally did not exceed that in ECFC–HCAEC and ECFC–HUVEC comparisons, suggesting minor yet significant differences between these endothelial cell populations. Gene set enrichment analysis and screening of pluripotency markers did not uncover notable differences; however, manual annotation of DEGs indicated that ECFC overexpress markers of all three endothelial differentiation lineages *(KDR*, *VWF*, *CD34*, *NRP2*, *FLT4*, and *LYVE1* in comparison with HCAEC; *NOTCH4*, *DLL2*, and *LYVE1* when compared with HUVEC).

Such an expression pattern assumes that ECFC are able to be differentiated into arterial, venous, or lymphatic endothelium, and might be closer to the latter lineage than HCAEC or HUVEC because of higher *LYVE1* expression. Moreover, ECFC have higher expression of venous endothelial marker *NRP2* than HCAEC, but also overexpress arterial differentiation marker *NOTCH4* as compared with HUVEC, suggestive of their intermediate specification between arterial and venous EC. This is corroborated by increased *COL1A1* expression by ECFC in comparison with HCAEC and HUVEC and augmented *COL1A2*, *COL4A1*, and *COL4A2* expression in ECFC compared with HCAEC, testifying to the upregulated production of extracellular matrix, including major components of the basement membrane. Importantly, proteomic profiling confirmed the results of RNA-seq, as levels of endothelial markers and angiogenesis molecules in ECFC were also intermediate between HCAEC and HUVEC. We therefore suggest PBMC-derived ECFC as a transitional cell population in the endothelial hierarchy (Figure 10).

Previous reports demonstrated that the global gene expression profile of ECFC largely depends on the source of their isolation and it varies between adult peripheral and cord blood [41], between the cultures grown in complete and serum-free medium (≈ 1100 DEGs) [42], and between blood collected from the patients with breast or renal cancer and healthy donors (342 and 71 DEGs, respectively) [43]. Intriguingly, ECFC isolated from breast and renal cancer patients shared a common 35-gene signature [43]. Cord-blood-derived ECFC from preterm neonates had > 700 DEGs in comparison with term newborns [44] while placental- and cord-blood-derived ECFC differed to a negligible extent (33 DEGs with fold change cutoff of 1.5) [30].

Co-culture of ECFC with mesenchymal stromal cells significantly alters their gene expression program (≈ 1700 DEGs) triggering endothelial-to-mesenchymal transition, potentially through elevated expression of *SNAI1* and *SNAI2* genes [45] encoding, respectively, transcription factors Snail and Slug, master regulators of this process [46,47,48]. Interestingly, in our study, *SNAI1* was upregulated whilst *SNAI2* was downregulated in ECFC compared with SAT-SVF, probably reflecting different stages of endothelial differentiation. Additionally, separate colonies within the single ECFC isolate may have distinct immunophenotype and transcriptomic signatures, e.g., arterial or lymphatic, and the extent of such heterogeneity (≈ 2250 DEGs) exceeded those obtained in our experiments (200–500 DEGs) and when blood and lymphatic ECFC were compared with primary mature dermal microvascular EC (≈ 1900 DEGs) [49]. A recent study proposed cytokine-like 1 (CYTL1), a protein promoting angiogenic sprouting, as a reliable marker of peripheral- and cord-blood-derived ECFC but not HUVEC or human saphenous vein EC being consistently > 10-fold overexpressed in ECFC [50]. This was partially replicated in our analysis as ECFC had 2.5- and 830-fold higher CYTL1 expression than HUVEC and SAT-SVF, respectively.

To the best of our knowledge, this study is the first attempt to conduct a transcriptome-wide comparison of ECFC to mature vascular EC populations such as HCAEC or HUVEC. We found that the global gene expression profile of ECFC was close to HCAEC or HUVEC, yet ECFC had transcriptomic signatures such as overexpressed common markers of all endothelial lineages and upregulated extracellular matrix genes. Transcriptome of ECFC drastically differed from SAT-SVF, attesting endothelial commitment of ECFC.

## 5. Conclusions

The gene expression profile and behavior of PBMC-derived ECFC show a sufficient extent of similarity to vascular EC for testing them in pre-endothelialization of bioartificial vascular grafts, whereas in terms of developmental biology they significantly differ from HCAEC and HUVEC in the expression of certain gene categories relevant for endothelial biology (endothelial specification markers and synthesis of extracellular matrix/basement membrane components).

## Figures and Tables

**Figure 1 cells-09-00876-f001:**
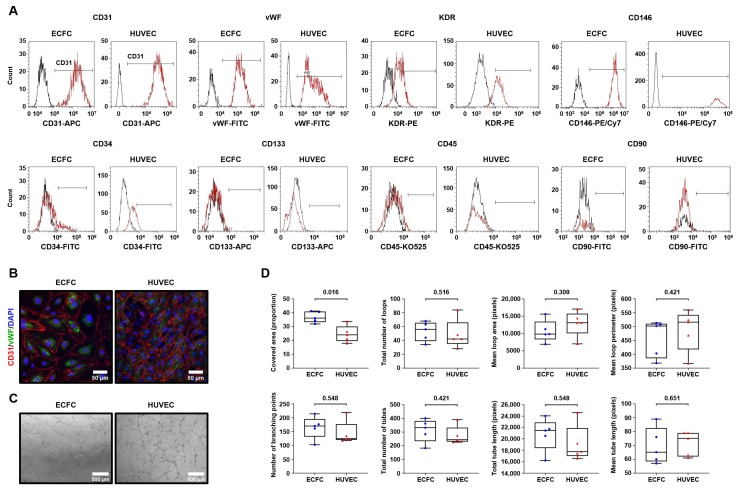
Peripheral blood mononuclear cells (PBMC)-derived endothelial colony-forming cells (ECFC) have characteristic endothelial immunophenotype and considerable tube-forming activity in Matrigel. (**A**) Flow cytometry immunophenotyping (representative graphs) found both ECFC and human umbilical vein endothelial cells (HUVEC) having CD31+vWF+KDR+CD146+CD34-CD133-CD45-CD90- expression signature suggestive of endothelial identity. Black color defines isotype control while red color is for specific staining; (**B**) Confocal microscopy (representative images) corroborates flow cytometry results demonstrating CD31-positive (red color, cell surface receptor) and vWF-positive (green color, Weibel–Palade bodies inside the cytosol) ECFC and HUVEC. Nuclear counterstaining was performed using 4′,6-diamidino-2-phenylindole (DAPI, blue color); (**C**) Phase contrast microscopy (representative images). Both ECFC and HUVEC have pronounced capability to assemble capillary-like tubes in Matrigel; (**D**) Semi-quantitative analysis of the tube-forming activity from the experiment in C. Each dot represents a representative image from one well of the culture plate (*n* = 5 wells per group). Whiskers indicate range, box bounds indicate 25th and 75th percentiles, center lines indicate median. *P* values are provided above the graphs in a numerical manner according to Mann–Whitney U test.

**Figure 2 cells-09-00876-f002:**
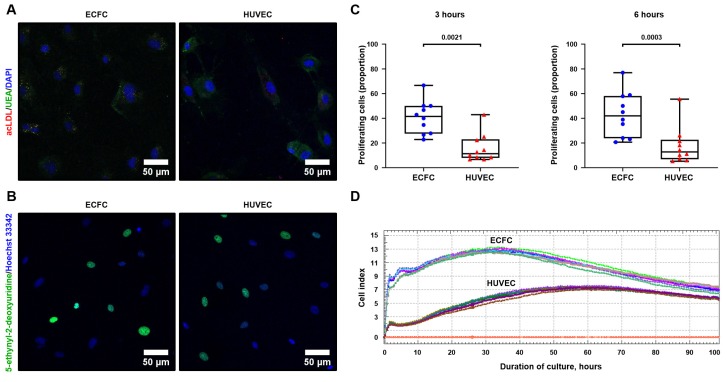
PBMC-derived ECFC internalize acetylated low-density lipoprotein cholesterol (acLDL), bind *Ulex Europaeus* agglutinin 1 (UEA), and exhibit high proliferation capacity. (**A**) Confocal microscopy (representative images) shows uptake of acLDL (red color inside the cytosol) and binding of UEA (green color around the cell membrane) by ECFC and HUVEC. Nuclear counterstaining was performed using DAPI (blue color); (**B**) A major proportion of ECFC and HUVEC are actively dividing as shown by 5-ethynyl-2-deoxyuridine incorporation into their nuclei (Hoechst 33342, blue). Representative images; (**C**) Semi-quantitative analysis of the proliferative activity from the experiment in B. Each dot represents one representative image from one well of the culture plate (*n* = 10 wells per group). Whiskers indicate range, box bounds indicate 25th and 75th percentiles, center lines indicate median. *P* values are provided above the graphs in a numerical manner according to Mann–Whitney U test; (**D**) Electrical impedance monitoring in ECFC and HUVEC cultures confirms the results from confocal microscopy. Proliferation capability was defined as cell index doubling time calculated automatically by the instrument software. Each line represents a result collected from one well of the culture plate (*n* = 7 per group). Red lines at the bottom indicate impedance of the blank solution (cell-free culture medium).

**Figure 3 cells-09-00876-f003:**
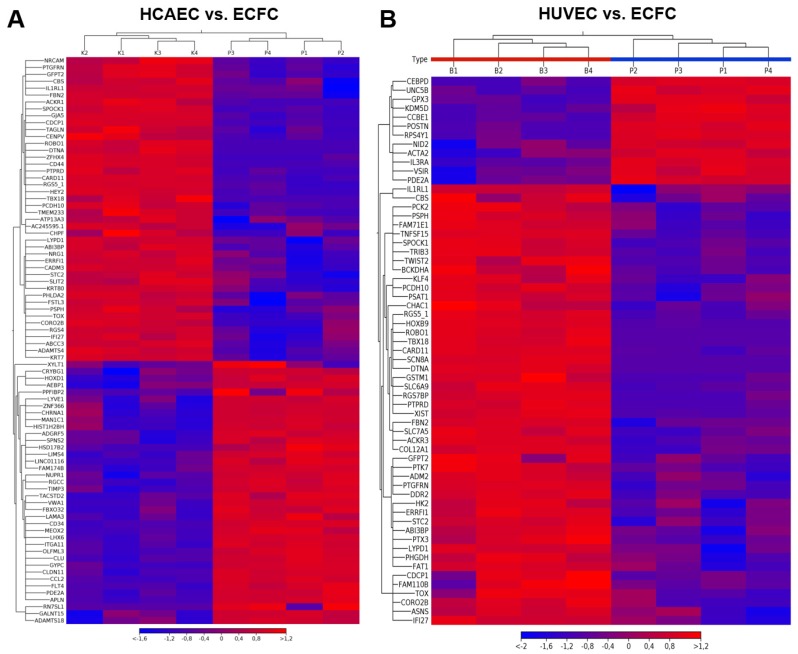
Heatmaps of differentially expressed genes between (**A**) human coronary artery endothelial cells (HCAEC) vs. ECFC (only those having fold change 0.1 (−10) or 10, respectively, and FDR *p*-value < 0.05 are represented in the heatmap); (**B**) HUVEC vs. ECFC (only those having fold change 0.2 (−5) or 5, respectively, and FDR *p*-value < 0.05 are represented in the heatmap). Top panel: dendrogram showing the results of hierarchical sample clustering. Left panel: Dendrogram showing the results of hierarchical differentially expressed genes (DEGs) clustering. DEGs labeling based on Ensembl annotation.

**Figure 4 cells-09-00876-f004:**
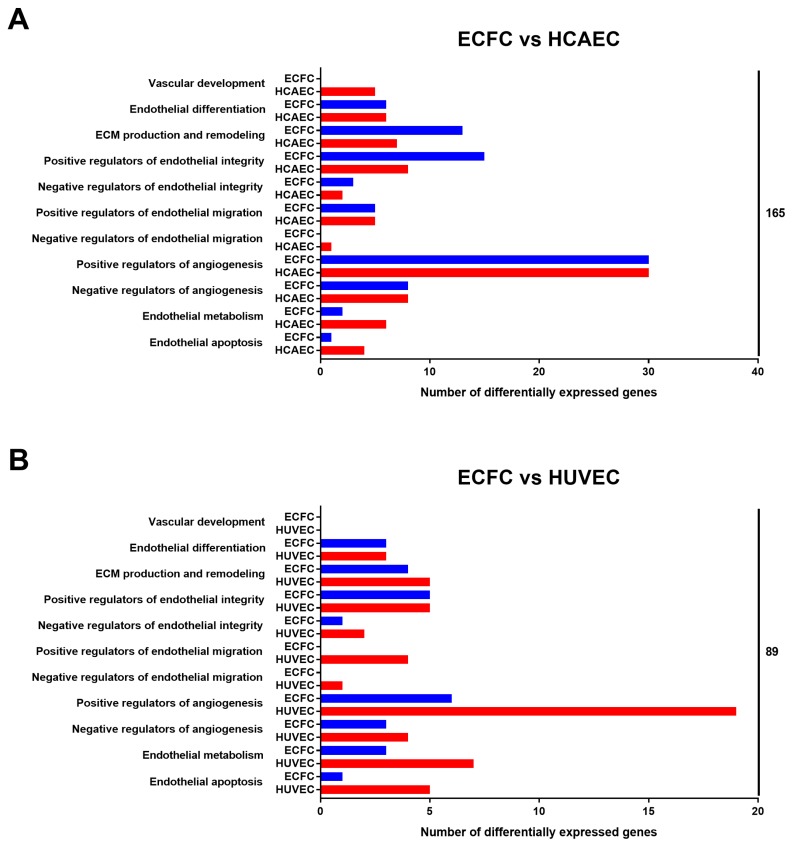
Pairwise comparison of (**A**) ECFC versus HCAEC and (**B**) ECFC versus HUVEC according to the manual annotation of DEGs (only endothelial biology-related categories shown). Total number of DEGs within the indicated categories is provided as the count at the right side of the graph.

**Figure 5 cells-09-00876-f005:**
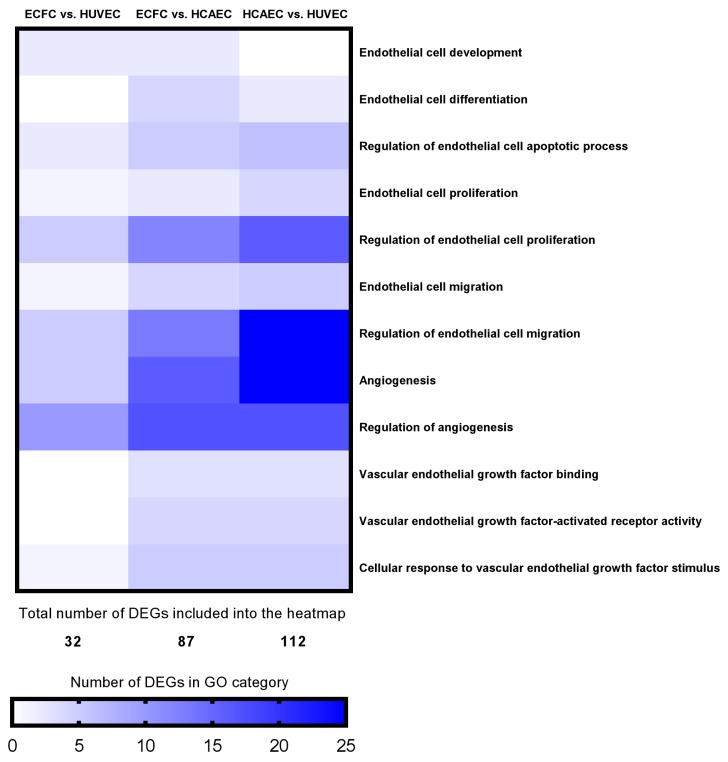
Heatmap representing the genes related to the regulation of endothelial homeostasis and angiogenesis and differentially expressed between ECFC, HCAEC, and HUVEC according to the GO functional enrichment analysis. Top panel: pairwise comparisons between the EC populations. Right panel: GO terms manually included in the heatmap. Bottom panel: a color mapping range.

**Figure 6 cells-09-00876-f006:**
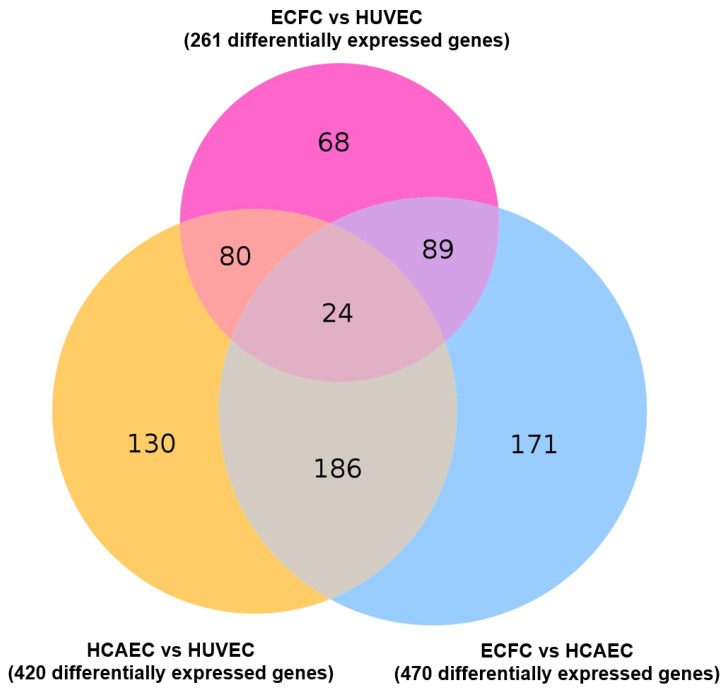
Venn diagram for the genes differentially expressed in ECFC, HUVEC and HCAEC.

**Figure 7 cells-09-00876-f007:**
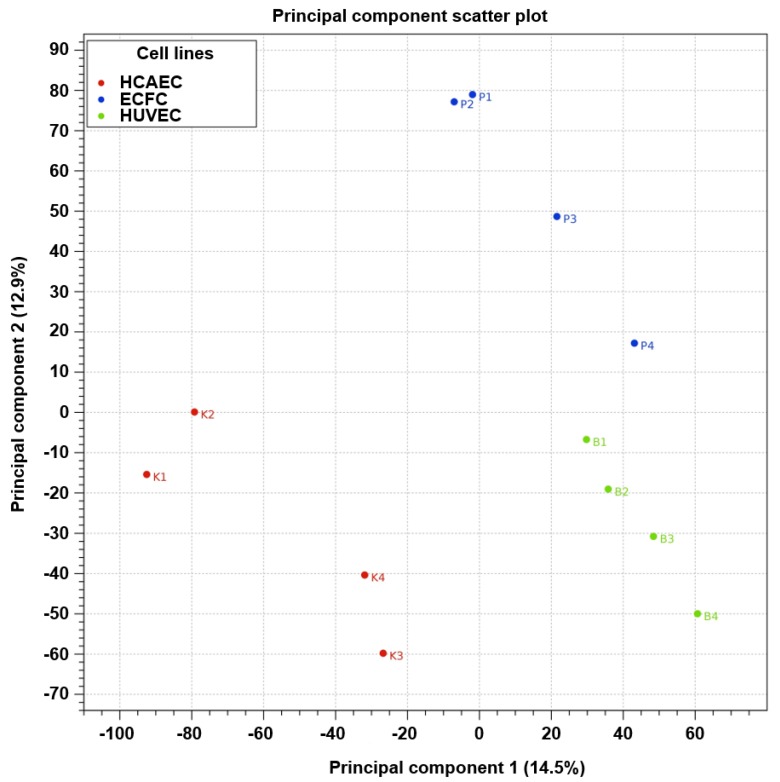
2D sample clustering based on a principal component analysis (PCA) of differentially expressed genes (DEGs). The red, blue, and green dots indicate HCAEC, ECFC, and HUVEC samples, respectively. Principal components are defined along the X and Y axes as component 1 and component 2, respectively; the proportion of explained data variance is indicated in brackets for each principal component.

**Figure 8 cells-09-00876-f008:**
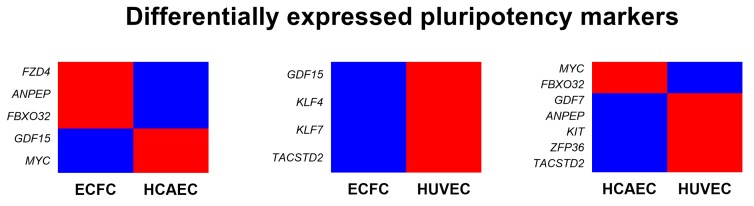
Heatmap showing the differential expression of pluripotency markers across ECFC, HCAEC, and HUVEC. Upregulated genes are colored red whereas downregulated genes are colored blue.

**Figure 9 cells-09-00876-f009:**
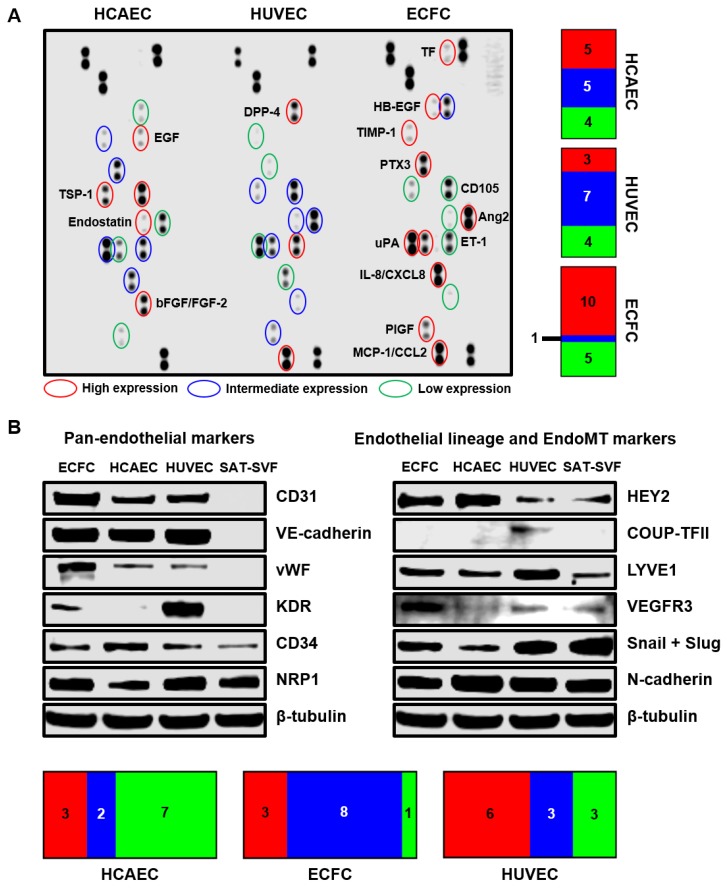
Proteomic profiling for ECFC, HCAEC and HUVEC. (**A**) Dot blot profiling for 55 angiogenesis-related proteins confirms high similarity between ECFC and mature vascular EC yet also indicates minor differences between these cell lines and generally validating RNA-seq results. Count to the right represents a quantitation of measured angiogenesis-related proteins in terms of their relative expression in HCAEC, HUVEC, and ECFC; (**B**) Western blotting for endothelial phenotype-related markers verifies endothelial identity of PBMC-derived ECFC and demonstrates an intermediate phenotype of PBMC-derived ECFC as compared to HCAEC and HUVEC in terms of HEY2, LYVE1, VEGFR3, Snail and Slug expression. β-tubulin was probed as a loading control. Count to the bottom represents a quantitation of measured endothelial phenotype-related markers in terms of their relative expression in HCAEC, HUVEC, and ECFC.

**Figure 10 cells-09-00876-f010:**
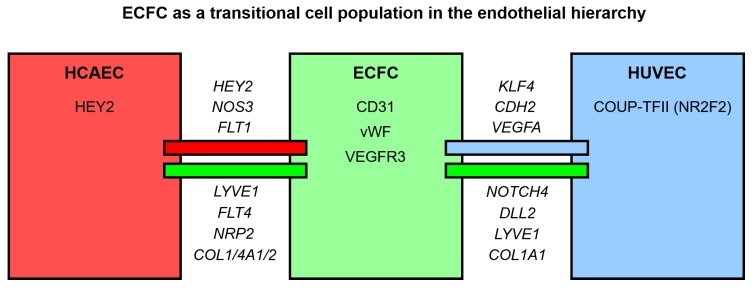
Graphical summary of the results. We suggest ECFC as a transitional cell population in the endothelial hierarchy overexpressing extracellular matrix/basement membrane components, lymphatic endothelial markers, and also venous endothelial markers as compared with mature arterial ECs and arterial endothelial markers in comparison with mature venous ECs. Changes at the protein level are within the boxes whereas changes at the transcriptomic level are above or below the lines (italicized gene names). Genes above the red and blue lines are upregulated in HCAEC and HUVEC, respectively, as compared with ECFC. Genes below the green lines are upregulated in ECFC in comparison with HCAEC (left green line) and HUVEC (right green line), respectively.

**Table 1 cells-09-00876-t001:** Median fluorescence intensities registered for different antigens and cell lines during flow cytometry analysis.

Antigen	ECFC		HUVEC		SAT-SVF	
	Isotype	Sample	Isotype	Sample	Isotype	Sample
CD34	3080	7155	1579	5220	3406	9438
CD309	4257	15,063	2272	13,974	1640	2155
CD146	5080	1,262,908	210	417,263	256	445
CD133	24,425	24,390	2083	1891	0	0
CD45	4128	4221	2353	2266	25,102	24,684
CD31	26,148	2,151,501	65	44,055	362	1213
vWF	4243	203,182	668	127,268	156,915	167,111
CD73	0	0	0	0	180	8235
CD90	4883	4915	4531	4598	1133	2934

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
