# Peer review of "Human Peripheral Blood-Derived Endothelial Colony-Forming Cells Are Highly Similar to Mature Vascular Endothelial Cells yet Demonstrate a Transitional Transcriptomic Signature"

_cells, 2020, doi:10.3390/cells9040876_

Round 1

Reviewer 1 Report

The authors have added protein expression data to the manuscript, which adds some novelty and provides support to the transcriptomic profiling results. However, the authors have failed to address the limitations pointed out below:

  • The authors fail to address the fact that comparison to SAT-SVF is quite irrelevant. Although they state that the manuscript does not emphasize the results as key point, ¼ of the results describes these findings and also abstract reports those results as major findings. The main focus needs to be in the characterization of the endothelial-like features and the limitations of comparison to SAT-SVF clearly stated.
  • None of the Reviewer-comments related to comparison to previous studies, presentation of pluripotency markers, or markers that would strengthen the conclusions that the ECFCs indeed represent “a transitional cell population” were addressed adequately. The number of differentially expressed genes is 261-420, what are the other genes that are not discussed? Which one of those are relevant to cellular differentiation? Please provide a schematic of the endothelial cell hierarchy that highlights the factors that support ECFCs as being intermediate cell population. Also the full list of the DEG’s should be provided in the supplement.
  • How was the list in Figure 4 generated as it only represents a sublist of total DEGs and not all DEGs as stated in the figure legend.
  • Line 304 says there are 700 DEGs between the ECFC and SAT-SVF and Figure 6A states 2,700 DEGs, why are the numbers different?
  • For section 3.3, it would be informative for the reader that all the genes relevant to differentiation, vascular development, endothelial development, VEGF-activated receptor activity, angiogenesis etc would be summarized in a heatmap (e.g. sublist of the current Figure 4, but representing ECFC, HUVEC and HCAEC in same heatmap).
  • The statement that “gene expression profile and behaviour of ECFC are sufficiently similar to vascular EC for using them for pre-endothelialisation of bioartificial vascular grafts” is merely speculative as no functional data is represented to support this claim.

Author Response

Please also see the attachment (we marked our response in red colour).

The authors have added protein expression data to the manuscript, which adds some novelty and provides support to the transcriptomic profiling results. However, the authors have failed to address the limitations pointed out below:

Major points

Reviewer: The authors fail to address the fact that comparison to SAT-SVF is quite irrelevant. Although they state that the manuscript does not emphasize the results as key point, ¼ of the results describes these findings and also abstract reports those results as major findings. The main focus needs to be in the characterization of the endothelial-like features and the limitations of comparison to SAT-SVF clearly stated.

Authors: We agree with the reviewer. We removed all mentions on SAT-SVF from the abstract and wrapped all results into one short paragraph (Revised Section 3.2) and also moved all Figures regarding SAT-SVF into the Supplement. The focus of the revised manuscript is clearly on the comparison of ECFC with HCAEC and HUVEC. We used SAT-SVF as a negative control population for the technical validation of the approach.

Reviewer: None of the Reviewer-comments related to comparison to previous studies, presentation of pluripotency markers, or markers that would strengthen the conclusions that the ECFCs indeed represent “a transitional cell population” were addressed adequately. The number of differentially expressed genes is 261-420, what are the other genes that are not discussed? Which one of those are relevant to cellular differentiation? Please provide a schematic of the endothelial cell hierarchy that highlights the factors that support ECFCs as being intermediate cell population. Also the full list of the DEG’s should be provided in the supplement.

Authors: We think that the results of manual annotation of DEGs (Section 3.3 and Revised Figure 4), heatmap analysis of GO terms related to endothelial homeostasis and angiogenesis (Revised Figure 5), Venn diagram based on the total DEG count (Revised Figure 6), and principal component analysis (Revised Figure 7) justify suggestion that ECFC represent a transitional cell population in the endothelial hierarchy. Although ECFC seem to be more similar to HUVEC than to HCAEC due to a smaller number of DEGs in Revised Figures 4 and 5 and lesser distance on principal component analysis (Revised Figure 7), they do overexpress the genes encoding extracellular matrix / basement membrane components, lymphatic endothelial markers, and also venous endothelial markers as compared with mature arterial ECs and arterial endothelial markers in comparison with mature venous ECs. In keeping with these findings, proteomics analysis demonstrated that ECFC are characterised by either highest or lowest expression of angiogenesis-related proteins in three-way comparison with HCAEC and HUVEC whereas both of these populations show a more uniform distribution of protein expression ranks (Revised Figure 8A). Intermediate expression of the key endothelial proteins (Revised Figure 8B) in ECFC included into the indicated three-way comparison also reinforces the results of transcriptome-wide analysis. We included this into the Discussion.

We provided a schematic of the endothelial cell hierarchy highlighting the factors that support our claim (Revised Figure 10). The full list of DEGs is provided in Supplementary Datasheets 1 (SAT-SVF vs. ECFC), 2 (ECFC vs. HCAEC), 3 (ECFC vs. HUVEC), and 5 (HCAEC vs. HUVEC).

We analysed a number of DEGs which have not been included into the text utilising a quantitative graph (manually annotated categories related to endothelial homeostasis and angiogenesis, Section 3.3 and Revised Figure 4) and a heatmap (GO terms related to endothelial homeostasis and angiogenesis, Section 3.3 and Revised Figure 5). Markers of endothelial specification are also extensively discussed through the manuscript; however, it is hardly technically possible to discuss all DEGs within the text. We re-screened the pluripotency markers and again found only scarce differences between ECFC, HCAEC, and HUVEC, representing the full results as a heatmap in Revised Figure 8. No differences have been found for SOX or ETS transcription factors. Previous studies are mentioned in the Discussion and the results are compared when possible.

Reviewer: How was the list in Figure 4 generated as it only represents a sublist of total DEGs and not all DEGs as stated in the figure legend.

Authors: Figure 4 (Figure 3 in the revised manuscript) was generated as follows:

Revised Figure 3A (HCAEC vs. ECFC): only those having fold change 0.1 (-10) or 10, respectively, and FDR p-value < 0.05 are represented in the heatmap.

Revised Figure 3B (HUVEC vs. ECFC): only those having fold change 0.2 (-5) or 5, respectively, and FDR p-value < 0.05 are represented in the heatmap.

Revised Figure S5 (SAT-SVF vs. ECFC): only those having fold change 0.001 (-1000) or 1000, respectively, and FDR p-value < 0.05 are represented in the heatmap.

This was done in a mentioned way to allow the gene names to be visible. If we include all DEGs into the heatmap, it would be technically impossible to read their names. Therefore, we selected the DEGs with a greatest fold change, and the cutoff has been chosen individually for each comparison.

Reviewer: Line 304 says there are 700 DEGs between the ECFC and SAT-SVF and Figure 6A states 2,700 DEGs, why are the numbers different?

Authors: The actual number of DEGs was 2,696 as stated in the text in relation to the Figure 6A (Figure S4 in the revised manuscript). 700 DEGs at line 304 was the number defined by a more stringent cutoff (fold change ≥ 10 and Bonferroni-corrected p-value < 0.05) to permit the manual annotation analysis. Yet, as the revised manuscript does not focus on SAT-SVF, we removed the statement on 700 DEGs from the revised manuscript, also making the analysis criteria uniform. Hence, the number of DEGs is 2,696, selected by the criteria applied to other comparisons (fold change ≥ 2 and FDR p-value < 0.05).

Reviewer: For section 3.3, it would be informative for the reader that all the genes relevant to differentiation, vascular development, endothelial development, VEGF-activated receptor activity, angiogenesis etc would be summarized in a heatmap (e.g. sublist of the current Figure 4, but representing ECFC, HUVEC and HCAEC in same heatmap).

Authors: We agree with the reviewer. We have drawn two additional figures (Figure 4 and Figure 5 in the revised manuscript) which respectively represent a quantitative graph and a heatmap demonstrating the number of DEGs within the suggested manually annotated categories (those related to endothelial homeostasis and angiogenesis, Figure 4) and GO terms (similar categories, Figure 5). We also summarised a total number of DEGs between ECFC, HCAEC, and HUVEC within these categories, thereby connecting functional enrichment analysis (Figures 4 and 5 in the revised manuscript) with a Venn diagram (Figure 6 in the revised manuscript). We also provided the data included into this analysis as a Supplementary Datasheet 4.

Reviewer: The statement that “gene expression profile and behaviour of ECFC are sufficiently similar to vascular EC for using them for pre-endothelialisation of bioartificial vascular grafts” is merely speculative as no functional data is represented to support this claim.

Authors: We agree with the reviewer and changed this statement in the Abstract to: “gene expression profile and behaviour of ECFC suggest their potential to be applied for a pre-endothelialisation of bioartificial vascular grafts”. This is a suggestion rather than a claim in such words.

We sincerely thank the reviewer for the valuable comments.

Reviewer 2 Report

Authors responded to all my comments.

Author Response

We sincerely thank the reviewer for the valuable comments.

Round 2

Reviewer 1 Report

The authors have addressed all my comments. Just few minor comments listed below:

-Please provide explanation for the colors in Figure 8.

-The Supplementary Datasheet 2 has filters that others do not have, please remove these for clarity.

-Please remove the nonsignificant rows (non-significant in all) from Figure 5 and 8 as they do not provide any relevant information

Author Response

We sincerely thank the reviewer for the valuable comments. Please also see the attachment if needed.

The authors have addressed all my comments. Just few minor comments listed below:

Reviewer: Please provide explanation for the colors in Figure 8.

Authors: We have reworked Figure 8 according to your suggestion below and provided the explanation for the colors (upregulated and downregulated genes are now coloured red and blue, respectively).

Reviewer: The Supplementary Datasheet 2 has filters that others do not have, please remove these for clarity.

Authors: We consistently removed all filters from Supplementary Datasheets to make them uniform.

Reviewer: Please remove the nonsignificant rows (non-significant in all) from Figure 5 and 8 as they do not provide any relevant information.

Authors: We removed the non-significant rows from Figures 5 and 8, respectively.

This manuscript is a resubmission of an earlier submission. The following is a list of the peer review reports and author responses from that submission.

Round 1

Reviewer 1 Report

The manuscript " Global gene expression profile of human peripheral blood-derived endothelial colony-forming cells is similar to coronary artery and umbilical vein endothelial cells” by Kutikhin et al compares the gene expression profile, immunophenotype, proliferation and tube formation capacity of human PBMC-derived ECFCs to HUVECs, HCAECs and SAT-SVFs. The data is interesting and questions pertinent. However, the study lacks in novelty and significant improvements in the presentation of results are needed.

Major points

The logic of using SAT-SVF as negative control is unclear and controversial. SAT-SVF represents heterogenous cells and surely provides a nice separation from ECFCs but the biological relevance is missing. They clearly do not represent endothelial-like cells. The study lacks novelty as a similar study has previously compared ECFCs from peripheral blood and endothelial cells isolated from adipose tissue and HUVECs (Ref 39) using similar methods. Therefore this study needs to make sure to provide additional insight to the field. I suggest the authors provide further comparison to existing data. How do the gene ontologies and pathways identified in this work compare to those from previous comparisons (Refs 39 for HUVEC vs ECFCs)? Are the results here in line with the expression profiles of peripheral blood derived ECFCs analyzed in previous studies (e.g. Ref 40)? The main finding of the paper is that ECFCs express markers of all endothelial lineages. Could the authors also provide heatmap looking at the expression of pluripotency markers, are those more highly expressed in their ECFCs? Whether this is a negative or a positive thing in vascular grafts should be discussed. How does the expression of lineage markers and pluripotency markers cluster the cell types by similarity (PCA/heatmap clustering)? What are the transcription factors that are predicted to be responsible for the differences (e.g. SOXs, ETS, ERG) and would the knockdown or overexpression of selected factors change the proliferative or angiogenic capacity of ECFCs? What are the other genes commonly differential between ECFCs vs HCAECs and HUVECs? Providing answers to there questions would significantly improve the sigfinicance of the paper.

Minor points related to M & M documentation

Please list the culture mediums used for HUVECs and HCAECs. Were the same medium and coating used for all cells before collection for gene expression analysis as this significantly affects the expression profile of cells and affects comparison? For Abs, please provide the Ab dilution used. Describe how many fields of image were analyzed using confocal microscopy for calculations. Please provide the number of tag counts per library information.

Reviewer 2 Report

This paper discuss of a very hot topic concerning endothelial heterogeneity and characterization of ECFC.

However, I have some major comments concerning data presented her and conclusion of these results.

Comparison with SAT-SVF and conclusion that ECFC and SAT-SVF are different are really expected. I don’t understand objectives to find different panel of gene expression since phenotype is so different. Comparison of HUVEC/HCAEC/ECFC is of interest. However RNAseq analysis show differences of expression and title of the paper is: "… is similar to …" I feel confuse about this. This comparison is the real important point of the paper Some difference are found: before discussing these differences confirmation of these results at protein level need to be done and functional studies realized to explore if expression in genes found here are at the origin of functional difference found on ECFC in contrast to mature cells. A recent paper found intracellular expression of CD133 in ECFC as a difference in contrast to HUVEC or HAEC: this paper need to be cited at least in discussion.

Reviewer 3 Report

In their manuscript entitled Global gene expression pofile of human peripheral blood-derived endothelial colony forming cells is similar to coronaryartey and umbilical vein endothelial cells, Kutikun et al. used RNAseq to compare the transcriptomic profiles of cells the classify as peripheral blood mononuclear cell endothelial colony forming cells (PBMC-ECFC), human coronary artery endothelial cells (HCAEC) and human umbilical vein endothelial cells (HUVEC) with the adipose tissue derive stromal vascular fraction (SAT-SVF). HCAEC and HUVEC were similar to PBMC-ECFC in terms of global gene expression, but differed from the stromal vascular fraction of adipose tissue. Before reaching this conclusion, it would be beneficial for the authors to provide additional information on the characterisation of the cell samples used. The main points are listed in the Comment below.

Comments

The paper is generally clearly written, although it was difficult to have a clear idea of the numbers of ECFC in each cell group and if the cells were derived from the same passage number. It would be helpful if the authors could provide information for the readers on the following: Peripheral blood (20 mL) was withdrawn from 8 male patients during percutaneous coronary intervention. How many white blood cells and post-Ficoll MNC were obtained from these patient samples? How many of these cells were cultured per 25cm2 flask? Do the authors assess clonal ECFC as described by Yoder and colleagues? If so, what was the absolute number of cells and PBMC-ECFC generated by day 19-22 of culture? Were the number of PBMC-ECFC quantitated by clonal growth using limiting dilution or sorting of single cells as described by Ingram et al. 2004? It is unclear how these PBMC-ECFCs relate to those described by Yoder et al., who quoted a frequency of 1 ECFC per 106 to 108 MNC sourced from normal donors. Do the authors have data comparing the numbers of PBMC-ECFC (HPP-ECFC and LPP-ECFC) from normal donors and donors undergoing percutaneous coronary intervention and that could be cited in the text of this manuscript? Ingram et al. in 2005 identified a novel hierarchy of ECFCs based on their clonogenic and proliferative potential, using HUVECs as the cell source. When they cultured the cells on collagen-coated dishes as adherent cells, some but not all of the colonies formed demonstrated high proliferative potential. These HPP-ECFCs were defined by their ability to clonally expand in vitro into more than 2,000 endothelial cells in 14 days and to be capable of replating into secondary HPP-ECFC with some of these secondary clones displaying the same capacity to form colonies of >2,000 cells in 14 days. These HUVEC derived ECFCs also had capacity to form new blood vessels in vitro and in vivo. HUVEC lose this ability with extensive passage. Could the authors please clarify which HUVEC passage they used, the number of cells plated per 25cm2 flask and the number of cells and clonal HPP-ECFC and LPP-ECFC generated by day 19-22 of culture and present in the samples used for their RNAseq? How do these numbers compare with those in PBMC-ECFC preparations from the same passage? Could the authors please provide similar information for the HCAEC as in (ii), although it is recognised that if these cells are commercially sourced then the initial growth and culture conditions may not have been provided by the company.? Have the authors compared the ability of the PBMC-ECFC, HCSEC and HUVEC to form vascular tubules in vivo? It would be of interest to know if these cell sources are equally efficient in this respect when used at similar passages.

Cell identification: The authors report that PBMC-ECFC and HUVEC have similar endothelial phenotypes identified as CD31+vWF+KDR+CD146+CD34-CD133-CD45-CD90-, form tubes in Matrigel over 16h, with PBMC-ECFC having higher proliferative potential. In Figure 1, HUVEC appear to be CD34+ in contrast to PBMC-ECFC. As the gating for positive vs negative cells in the FACS histograms seems quite arbitrary, it would be more accurate if the authors quoted the Median Fluorescence intensities vs those of the negative controls. Page 4. Proliferation was measured over approx. 4 days and used to calculate doubling time. From studies of Yoder and colleagues, it would seem unusual for ECFC in the HUVEC preparation to have a lower proliferative potential when compared to the PBMC-ECFC from the adult patients. The absolute values of ECFC in each sample at the same passage would strengthen the results presented here and provide confidence that cells at similar passages were being compared. On Page 4 and Figure 2, it would appear that Acetylated low-density lipoprotein cholesterol (acLDL) uptake and Ulex Europaeus agglutinin 1 (UEA) binding are exclusive to ECFC or endothelial cells. These characteristics are shared with myeloid proangiogenic cells (see papers by MC Yoder and colleagues) and hence this description needs clarification. On page 9, CD34 is referred to as an endothelial cell marker. While this antigen marks endothelial cells, it is also found on hematopoietic stem/progenitor cells, on proangiogenic cells and on some stromal cells. In the text, the authors describe SAT-SVF as CD31-vWF-KDR-CD146-CD34+CD73+CD45-CD90-. However on page 7 Figure 3, these cells appear to be CD31+, and for at least some of these cells to be CD90+ and CD146+. As the gating for positive vs negative cells in the FACS histograms seems quite arbitrary here also, it would be more accurate if the authors quoted the Median Fluorescence intensities vs those of the negative controls.